**Data Availability Statement:** All relevant data are within the article and its Supporting Information files.

# "I'm not gonna be able to do anything about it, then what's the point?": A broad group of stakeholders identify barriers and facilitators to HCV testing in a Massachusetts jail

**Alysse G. Wurcel**[1,2]*, Jessica Reyes[2], Julia Zubiago[1], Peter J. Koutoujian[3], Deirdre Burke[1], Tamsin A. Knox[2,4†], Thomas Concannon[4], Stephenie C. Lemon[5], John B. Wong[2,4], Karen M. Freund[2,4,6], Curt G. Beckwith[7], Amy M. LeClair[4,6]

1 Division of Geographic Medicine and Infectious Diseases, Department of Medicine, Tufts Medical Center, Boston, MA, United States of America, 2 Tufts University School of Medicine, Boston, MA, United States of America, 3 Middlesex Sheriff's Office, Medford, MA, United States of America, 4 Department of Medicine, Tufts Medical Center, Boston, MA, United States of America, 5 Department of Population and Quantitative Health Sciences, University of Massachusetts Medical School, Worcester, MA, United States of America, 6 Department of Medicine, Institute for Clinical Research and Health Policy Studies, Tufts Medical Center, Boston, MA, United States of America, 7 Division of Infectious Diseases, Alpert Medical School of Brown University, The Miriam Hospital, Providence, Rhode Island, United States of America

† Deceased.
* awurcel@tuftsmedicalcenter.org

## Abstract

### Background

Despite national guidelines promoting hepatitis C virus (HCV) testing in prisons, there is substantial heterogeneity on the implementation of HCV testing in jails. We sought to better understand barriers and opportunities for HCV testing by interviewing a broad group of stakeholders involved in HCV testing and treatment policies and procedures in Massachusetts jails.

### Methods

We conducted semi-structured interviews with people incarcerated in Middlesex County Jail (North Billerica, MA), clinicians working in jail and community settings, corrections administrators, and representatives from public health, government, and industry between November 2018—April 2019.

### Results

51/120 (42%) of people agreed to be interviewed including 21 incarcerated men (mean age 32 [IQR 25, 39], 60% non-White). Themes that emerged from these interviews included gaps in knowledge about HCV testing and treatment opportunities in jail, the impact of captivity and transience, and interest in improving linkage to HCV care after release. Many stakeholders discussed stigma around HCV infection as a factor in reluctance to provide HCV testing or treatment in the jail setting. Some stakeholders expressed that stigma often led decisionmakers to estimate a lower "worth" of incarcerated individuals living with HCV and therefore to decide against paying for HCV testing.".

**Funding:** This work was supported by the National Institutes of Health, grants 1KL2TR002545-01 (to AGW), K08HS026008-01 (to AGW), P30AI42853 (to CB), and R25DA037190 (to CB).

**Competing interests:** The authors have declared that no competing interests exist.

## Conclusion

All stakeholders agreed that HCV in the jail setting is a public health issue that needs to be addressed. Exploring stakeholders' many ideas about how HCV testing and treatment can be approached is the first step in developing feasible and acceptable strategies.

## Introduction

Advances in hepatitis C virus (HCV) treatment make worldwide eradication an achievable goal [1], but eradication of HCV will require increased HCV testing in prisons and jails [2, 3]. One-third of United States (U.S.) inmates have HCV [4], far exceeding the 1–2% prevalence in the general U.S. population [5], but <1% of inmates with HCV have received treatment [6].

Extensive modeling has shown that HCV testing in people who are incarcerated prevents community spread of HCV and is cost-effective [3, 6, 7]. The joint Infectious Diseases Society of America and American Association for the Study of Liver Diseases guidance statement recommends HCV testing in all people who are incarcerated, and the Federal Bureau of Prisons endorsed opt-out testing [8, 9]. The vast majority of research on implementation of HCV testing in the criminal justice system has focused on prisons, where people are incarcerated for longer periods of time [6, 10, 11]. No guidelines exist for HCV testing in jails, where most incarcerations last less than 6 months. Increasing HCV testing in jails [12] and linking to HCV treatment on release have been shown to be feasible and effective [13–15], but only 4% of jails offer routine opt-in HCV testing [16].

Nationally, several barriers to HCV testing have been discussed, but little research has documented the views of different stakeholders involved in HCV testing and treatment decisions [17, 18]. Understanding the experiences and opinions of key stakeholders—including people who are incarcerated, corrections' administrations, public health representatives, and clinicians—is a crucial initial step to improving HCV testing access in jails. The goal of this study was to use a grounded theory approach to assess perspectives of key stakeholders about the current state of HCV testing as it relates to the provision of HCV treatment during incarceration in jails.

## Methods

The Tufts Health and Sciences IRB approved this study (#13054). We recruited individuals to participate in twenty-minute semi-structured interviews, using the "7-Ps" taxonomy to design a purposive sampling strategy [19]. Participants were sampled from the following groups: Patients & the Public, Providers, Payers, Purchasers, Policy Makers, Product Makers, and Principal Investigators. We recruited all detainees and corrections' staff from Middlesex County jail in North Billerica, Massachusetts, one of the largest jails in Massachusetts. Middlesex County jail offers HCV testing to all detainees during the nursing intake process, which occurs soon after incarceration. HCV testing can be requested by incarcerated individuals at any time, either in person with clinicians or by submitting a hand-written request to medical providers.

### Recruitment, enrollment and interviews with incarcerated participants

People who were incarcerated made up the patient sample, and they were recruited during two weekly orientations (when new detainees are instructed about the processes and policies of the

jail) at the Middlesex County Jail. At the end of these orientations, research assistants (JR, JZ, or DB) described the study and asked if anyone would be interested in participating. Detainees interested in the research approached the researchers, who wrote down their names. The researchers then went to a private room, and asked the corrections officer to bring in people who had their name on the list, one by one. The corrections officer was not in the room during the interview. Research staff emphasized multiple times during the recruitment process, consent, and before participating in the interview, that participation was voluntary, and there would be no repercussions if the person decided not to participate. The study details were explained and there were opportunities for detainees to ask questions prior to giving their verbal consent. After consent was given, the study staff signed the consent form with the date and the time signifying that they received verbal consent from the participant.

## Recruitment, enrollment and interviews with non-incarcerated participants

The Middlesex County Jail staff, including the clinicians and corrections administrators, were approached for participation via an email describing the study. If the recipient responded with interest, the researcher scheduled a time to meet with them privately at the jail. Community clinicians, public health and policy representatives, and people employed by pharmaceutical companies were identified and recruited as follows. Those known to the research team were contacted via email describing the study, and interviews were scheduled at their convenience and in their preferred location To facilitate adequate recruitment, all non-incarcerated stakeholders were asked to identify other individuals who would have insight on the topic being studied. This snowball sampling continued until representatives from each of the "P" groups had participated [20].

The study was explained prior to consent. The interviewer (JR, JZ, or DB) began with an icebreaker (see S1–S3 Files). Table 1 indicates the role of the participant and the "P" in which they were categorized. For readability and anonymity, we will refer to participants by their "P" group, with patients referring to the participants who were incarcerated.

## Data collection

We developed two 30–45 minute semi-structured interview guides: one for people incarcerated in jail and another for other stakeholders. Both guides sought to explore approaches to improving the system of HCV care in the jail setting. These were pilot-tested by a research

**Table 1. 7 Ps: Study participation.**

| Stakeholder Category | Study Specific Participant | Participants In Study | Participants Approached | Acceptance Rate |
|---|---|---|---|---|
| Patients | People who are incarcerated | 21 | 67 | 31% |
| Provider | Clinicians (total) | 9 | 15 | 60% |
| | In jail | 4 | 6 | 66% |
| | In Community | 5 | 9 | 55% |
| Purchasers & Payers* | People who oversee jail operations | 7 | 12 | 58% |
| Policy Maker | Public health, policy and government employees | 9 | 14 | 64% |
| Product Maker | Pharmaceutical industry representatives | 2 | 7 | 29% |
| PI | Researchers | 2 | 5 | 40% |
| Total | | 50 | 120 | 42% |

*(in text, will be referred to as "purchaser" for readability).

assistant (JR) in Tufts Medical Center's Infectious Disease Clinic with patients who had previously experienced incarceration. The interview guide for people who were incarcerated focused on personal knowledge or lived experience with HCV and their perspectives on access to and quality of healthcare in the jails. The interview guide for people who were not incarcerated focused on access to HCV care, quality of healthcare, and payment for HCV care in the jail setting.

Three research assistants (two white women and one Black-Hispanic woman) trained in qualitative data collection (JR, JZ, DB) conducted the interviews. Regulations did not permit recording devices inside the jail, therefore interviewers worked in teams of two, one transcribing and the other conducting the interview and taking notes. Neither the incarcerated individuals nor the participants who worked in the jail were allowed to accept incentives. Interviews with participants outside of jail were audio-recorded (with the interviewee's permission) and professionally transcribed. The interviewer took notes as well. These participants were compensated with a gift card ($50). Interviews were completed in English or Spanish, depending on preference of the participant.

### Data analysis

Transcripts were analyzed using Dedoose 6.1.18, a web-based software program. After developing a preliminary deductive codebook for each of the two interview guides [21], the research team coded a subsample of interviews (JR, JZ, AGW) and revised the codebook to include any and all emergent themes [20]. This iterative process continued until the team had produced an exhaustive list of codes (see codebook). We analyzed data thematically, where frequency and co-occurrence of codes were used to collapse and expand codes iteratively until a core group of key themes emerged. We resolved discrepancies in coding using a comparison and consensus approach. After completing coding of both sets of data, the research team compared the themes from each to determine overlaps or points of divergence to highlight in the results. Analysis revealed that we achieved thematic saturation with both samples, suggesting that interviewing more participants would not have provided additional novel information.

## Results

Fifty people (42% of people approached) agreed to be in the study, including 21 people who were incarcerated (Table 1). Below we discuss themes that emerged from the interviews. Participants were asked to self-report their age, race/ethnicity, and gender (Table 2).

### Knowledge about HCV testing policies

Knowledge about HCV testing opportunities in jails varied, with several incarcerated people unaware that HCV testing is available on request in the jail. Of the 21 incarcerated patients interviewed, few (2/21) recalled being asked if they wanted HCV testing, and more than half (13/21) said they were not offered testing. One such patient said, "I don't think I've ever been offered testing. It's kind of weird that no one has asked. I probably should, just for the hell of it. . . It would be good to know." Another reported only being tested for HCV in the jail, "I don't know where I would go [for testing] if I were home. I've only been tested in jail." A third echoed these themes, saying he may have been offered the test 5 years ago in jail, but he was not sure. He continued, saying even if they asked him to have the HCV test, he wouldn't be interested because he needed to "learn" more about it. He said, "They need to tell me 'it's for this and that.' They need to tell me about the complications." There was also variability in clinician response, as one provider who worked in jail reported that HCV testing was not offered to everyone, while others said that it was offered at intake or sick call.

**Table 2. Participant demographic information.**

| Incarcerated Participants | | |
|---|---|---|
| **Gender Identity** | Male | 20 (95%) |
| **Age** | 18–24 | 5 (24%) |
| | 25–34 | 9 (43%) |
| | 35–44 | 3 (14%) |
| | 45–54 | 4 (19%) |
| **Race/Ethnicity** | White/Non-Hispanic | 8 (38%) |
| | White/Hispanic | 1 (5%) |
| | Black/Non-Hispanic | 4 (19%) |
| | Black/Hispanic | 2 (10%) |
| | American Indian | 1 (5%) |
| | Asian | 1 (5%) |
| | Other/Hispanic | 4 (19%) |
| Non-Incarcerated Participants | | |
| **Gender Identity** | Male | 15 (52%) |
| **Age** | 25–34 | 5 (17%) |
| | 35–44 | 5 (17%) |
| | 45–54 | 4 (14%) |
| | 55–64 | 5 (17%) |
| | 65+ | 2 (7%) |
| | Missing | 9 (31%) |
| **Race/Ethnicity** | White/Non-Hispanic | 23 (79%) |
| | Black/Non-Hispanic | 1 (3%) |
| | Black/Hispanic | 0 (0%) |
| | Asian | 2 (7%) |
| | No answer given | 3 (10%) |

While many patients who were incarcerated had heard about HCV or heard rumors about other people having it, others were less informed. Those patients with less knowledge of HCV expressed more mistrust of the healthcare system and less interest in testing: "I heard [HCV] was something I don't want. I was told it was an STD, but honestly I don't know. I hear a lot of shit about shit and don't believe it." Several incarcerated patients felt HCV was primarily transmitted through sex, and many did not think HCV could be cured. One patient recalled, "I was tested for [HCV] in 2017. It was painless. I didn't feel like I had control. I think they were testing everyone for it. It's something you had to do."

## Stigma and deservingness

Opinions on the importance of stigma differed. Several participants (incarcerated and non-incarcerated) felt that HCV was not a stigmatized illness, including one policy maker who works in the jail, "years ago, when it came to HIV, people were reluctant to participate in HIV testing. It's not the same with Hep C. There is no stigma surrounding Hep C and inmates." One incarcerated patient said, "HIV is serious. Hepatitis C isn't as serious. I have an uncle that died of HIV." However, other participants discussed how stigma was a real barrier to getting tested and treated for HCV. One provider who was in recovery said, "Addicts are not, in their mind, good people, they're weak, stupid, evil, mean, selfish... you know all those things. Which, unfortunately, many addicts have heard that so many times, that they start internalizing that." A provider described the process of drawing blood in jail as negative with "phlebotomists who are

very judgy and unkind and blame the patient when they can't get blood from them [because of vein damage due to IV drug use]" or "really judgmental remarks. . . usually around drug use." A product maker/pharmaceutical representative brought up that if a person who is incarcerated requested HCV testing, they might be targeted: "So, you know sex or doing drugs or sharing needles within the prison system. . . people are much more reluctant to ask for a test if they feel like they might get in trouble because when they say, 'well, why do you need a test?'"

Many participants discussed the stigma around incarceration as impacting whether they deserved HCV care. One incarcerated patient said, "In jail they wouldn't treat [HCV] as serious because people look down on inmates. . .. A white coat will offer more help than an officer, that's my opinion. I'm a human like everyone else." Another said, "Human beings deserve to be treated. They deserve to get treatment and move on." When asked about what should be done if someone with HCV gets treatment, clears the illness and then is reinfected, a third incarcerated patient said "ew, if you got it twice you are just nasty," but then said, "I think they should get treated again- because that's kind of right. . . almost nobody deserves to have that or any disease."

Several participants commented on how HCV was treated differently than other diseases. When asked who pays for HCV treatment for people in jail, one incarcerated participant said, "I don't know. It's not like the Cancer Treatment of America. You don't hear about that." A purchaser said, "So when they think about the cost of one course of sofosbuvir the way that they're thinking about it is, 'Okay, this Puerto Rican is costing us 16 grand,' as opposed to, 'Hey, we're treating our inmate population and we're eradicating this horrible disease.'" A policy maker said, "I say to folks, 'Guys, every medication is expensive.' That's just the way that it is, you know. But we don't hear about the cost of those other medications because somehow that illness is more socially acceptable."

## Impact of captivity and transience on HCV testing

Several participants (incarcerated and non-incarcerated) felt that jail was a good time to offer HCV testing, often referring to other tests they do in the jail like testing for HIV and tuberculosis, including a payer who worked in jail and said "If you come here, get tests, the test is confirmed. . . if you detox and re-evaluate life. . . it's a good time in a sense that we have a captive audience." The period of "sobriety" in jail was highlighted as an opportune time to initiate treatment.

The role of the jail as the sole healthcare provider for people who are incarcerated emerged frequently among the non-incarcerated stakeholders. A payer noted "The only access to healthcare that a lot of our indigent populations and substance abusers who are at the highest risk of being infected with hepatitis C. . . oftentimes is only when they're in jail so I feel [testing for HCV in jail] is a necessity." Transience was discussed several times in interviews as a barrier to HCV care, including one purchaser working in jail who said, "This is a transient population, it can be hard to follow treatment from start to finish since it takes 8–12 weeks." A policy maker felt jail was the appropriate time to treat, since "It's not that they're transient in their time in jail, their entire lives are transient. . . You should treat as much as you can while you can."

## Impact of HCV testing on limited budgets

Although HCV should be treated "like any other ailment. . . like HIV, liver disease or cancer. . . the thing about Hep C treatment is that it's expensive. . . jails have a fixed budget. 80% is fixed cost. . . payroll, food, keeping the lights on," according to one payer. Another payer discussed the many complexities associated with HCV treatment in jails, from budgeting to logistics to

staffing, and "how we figure out whose responsibility it is to make all of these things fall into place, so it's a challenge and concern." A policy maker highlighted the economic barrier of HCV testing and subsequent treatment in the jails, (which house people short-term), in comparison to the prisons who have long-term healthcare responsibilities, saying HCV treatment "is not only beneficial but it's cost effective. The problem is cost effective for whom. . . if you're only responsible for that person's care for six months, a year, two years, three years, and then somebody else is going to be responsible for the cost of not doing something now, then it's not cost-effective for you. It's cost-effective for society. But not for you."

Participants commented on cost of HCV treatment in the jail setting. One provider proposed that "health insurance should pay. MassHealth should pay too, either through taxes or the state budget." A policy maker said "As soon as somebody becomes incarcerated, they're the responsibility of the state. If that person is then released more than likely they will be on some MassHealth plan. . . if the state is going to pay for it one way or the other, why not make sure somebody's starting, completing and getting all the proper follow-up." The role of state government in financing HIV medication was discussed as an example by a purchaser: "I remember when we talked about. . . some of the medications that can make an impact, positive impact on people who have HIV. . . we looked at the cost of it and it was just staggering, but the state stepped up and gave us reimbursement for the HIV medication and it became just another medication that we gave."

## The role of linkage to care programs

One of the providers explained that they understood why providers in jail may be reluctant to test: "If you're in a situation where you're the only provider taking care of a lot of patients than it may be well, 'okay if I test and find this [and] I'm not gonna be able to do anything about it, then what's the point?'"

Linkage to care post-release was suggested as an opportunity to maximize testing during incarceration, and relieve the burden of treatment for corrections staff. Incarcerated patients were interested in linkage to HCV care after release: "I want to find out how to get rid of [HCV]. . . I'll only be here a week or so, I'll do it outside." Another reported "There was no follow up after jail. They could have. . . told me my options, of what could have been done, instead of just saying I could get it treated told me how to go about getting treated, letting me know more about the treatment." People who worked in jails agreed, saying "We link people with opioid use disorder to care. Now we do the same thing for mental health. But we're not doing as good a job with people who have health issues. I think it would be good to introduce and link people to providers [on the outside]." One policy maker envisioned it as "drilling down to the level of getting someone an appointment, right. Here's your first appointment at your provider in the part of the Commonwealth that you're going to. That would make sense. You know, and just have as close to a warm handoff as you could."

Additionally, improving communication between jails and the community was highlighted by providers because people working in the jails have limited access to prior testing, and sites on the outside may not receive healthcare information from the jails.

## Discussion

In this study, we highlight the voices of several perspectives involved in making decisions for themselves or others about HCV testing and treatment in jail. There was near universal agreement that people incarcerated in jail often face barriers to accessing healthcare in the community, and the criminal justice system provides an opportunity to begin the cascade of care by testing for HCV. There was less consensus among stakeholders about the role of the jail in

providing HCV treatment, especially given fixed healthcare budgets. Some stakeholders felt that offering testing without treatment was pointless given the difficulty of coordinating continuity of care following release, however other stakeholders felt that HCV testing with linkage to HCV treatment after re-entry into the community was an acceptable alternative.

Our research builds on previous work done nationally and internationally to understand barriers and facilitators to HCV testing in incarcerated settings. The vast majority of research has focused on the views of people who are incarcerated in prisons, not jails. The prison research has highlighted stigma and lack of knowledge as powerful deterrents preventing HCV testing [22–25]. In our study, stakeholders from all different perspectives compared HCV to other "socially-acceptable" illnesses, and the topic of deservingness—both of the infection as punishment and the treatment as reward—were themes that emerged. Overall, participants in our study who were incarcerated at the time of their interview were interested and motivated to learn more about HCV testing and treatment, similar to the findings of previously published studies [26, 27]. While Middlesex County Jail offers HCV testing on request, one incarcerated participant in our study felt he did not have the autonomy to refuse HCV testing, a theme that previously emerged in qualitative interviews with incarcerated populations in California [28] and currently highlights the complexities of explaining optional testing in criminal-legal settings.

The feasibility and cost-effectiveness of HCV testing in jails—especially opt-out testing—has been demonstrated in several different state jails, including Massachusetts [13, 27, 29, 30]. Testing is most often offered at intake, although some places do infectious diseases "exit" testing before people return to the community [31, 32]. Notably, in our study the majority of incarcerated people that we interviewed did not recall being offered HCV testing. The early days of incarceration can be particularly challenging as people deal with withdrawal symptoms from substances, emotional struggles/fear with incarceration itself, and adjust to the setting [33]. Offering exit testing could be a useful strategy to optimize uptake, though there are challenges to delivering results prior to re-entry into the community.

A major strength of our study is that we built collaborations with correctional administrators, facilitating access to important and under recorded perspectives. The opinions of sheriffs, clinicians, and administrators working in the criminal justice system are also key to improving HCV care. Outside of the work done in Ireland and England [24, 34], there is little published on their perspectives of barriers and facilitators to providing adequate HCV care in correctional settings. Given the presence of several high-profile lawsuits related to HCV care provision in jails [35], open, nonjudgmental communication is critical to understanding the decisions about HCV testing and treatment made by those working in the criminal justice system.

The importance of engaging people who have been incarcerated to help develop HCV testing strategies cannot be overstated. Lived experience should inform discussions about the best way to communicate preventative healthcare strategies with incarcerated people often who have low health literacy and may lack trust in clinicians working in the jail. Involving people who are incarcerated in discussions about their care is a critical step towards developing HIV/HCV care strategies that work for them [24, 25, 31, 33]. Engaging people who are incarcerated in the development of programs for them has been shown to improve effectiveness in programs for transgender detainees [36] and STI testing for women who are incarcerated [37], as well as understanding perceptions of coercion in mental health courts [38].

For public health officials and clinicians working in the community, incarceration represents an important public health opportunity for HCV testing and treatment. The Middlesex Sherriff's Office believes that HCV testing and treatment are medical decisions that should be made between qualified healthcare professionals and their patients. The short period of time

that people are incarcerated, however, makes operationalization of HCV testing programs tricky. In communications with the Middlesex Sheriff's Office, the median length of detention for the pretrial population in Middlesex County Jail—about 50% of the jail population—was 13 days [IQR: 2, 50]. Juggling the complex medical issues of inmates (e.g. diabetes and HIV), finite phlebotomy services, and limited space can present challenges to the coordination of seemingly easy one-time blood draw for HCV testing. Although HCV treatment has been iteratively simplified—now with 8–12 weeks of treatment for all genotypes achieving >95% cure —the process of ordering HCV medications from the pharmacy, reviewing interactions, counseling the patient, and delivering the treatment can often seem an insurmountable task when people are incarcerated for short periods of time. Working within these constraints, many jails have implemented innovative HCV testing strategies focusing on linkage to care in the community.

Robust evidence supports the feasibility of HCV treatment in jails, but medication costs remain a barrier. Modeling from a Rhode Island (RI) study estimated that treating all sentenced people for HCV would cost $17 million dollars, about six times the entire pharmacy budget for the RI prisons [39]. There are several strategies to address the cost of HCV treatment, including generic competition, voluntary licensing, and tiered pricing [40, 41]. Louisiana has led the country in innovation by gaining CMS approval for a new payment model to treat HCV in the corrections and Medicaid program [42]. Linkage to care in the community is another option to address costs. Even when people are released mid-treatment, linkage to care post-release with continued treatment has resulted in rates of HCV clearance on par with non-incarcerated populations [15]. Of note, recent "real-world" studies suggest that successful treatment of HCV is possible with shorter courses of treatment that minimizes risk of treatment interruptions or sub-optimal adherence [43, 44], e.g., with 8 weeks required for most HCV regimens. The risk of treatment interruption could be avoided by planning to provide HCV medications on release with assurance of linkage to community-based HCV care. Progress in HCV testing and treatment in carceral settings needs to be supported by outreach in the community to ensure continuity of care, and access to health insurance post-release.

This study informs recommendations for advancing HCV-related healthcare. HCV is common in jails, but stigma persists. No matter how much education is provided to encourage HCV testing, if people who are incarcerated are fearful of denigrations while getting blood drawn, then that creates a major barrier. The culture of healthcare in jail has reformed substantially, with heightened focus on addiction as a disease, but there is still work that needs to be done to get every employee working with the same approach. Feedback from people who are incarcerated is crucial to improve the system of care delivery, even if it is simple impromptu discussions asking people to recall what questions were or were not asked during nursing intake. Although nursing intake is an opportunity to introduce testing, the emotional strain may inhibit full engagement among people who are newly detained, therefore alternative times for approaching and encouraging testing should occur. Linkage to care post-incarceration could facilitate HCV treatment to people who may not be in jail for a long enough time period to complete treatment. This could be coupled with other referrals such as linkage to substance use disorder treatment, housing, or job services programs. Finally, regarding the cost of HCV medications, legislation at the state and federal level should extend insurance coverage to people in jail during the pre-sentencing period, and reinstate insurance in the thirty days prior to release, thus reducing the cost of HCV medications for jails and support continuity of their HCV treatment and overall healthcare [45, 46].

Our study has the following limitations. The first limitation is that we interviewed people in only one jail in Massachusetts, that housed men (although one interviewed person identified as female). Second, interviewers were all female (one was Afro-Latina and two were white),

often discordant with interviewee demographics, potentially biasing their answers [47]. Finally, social desirability bias may have impacted interviewee responses. For participants who were incarcerated, two people observed their responses, potentially adding additional pressure to respond in a certain way. For the non-incarcerated stakeholders, the presence of a tape recorder may have exacerbated response bias.

## Public health implications

Improving access to HCV care in jails is crucial to end the HCV epidemic and cannot be done without broad and inclusive stakeholder engagement. Stakeholders interviewed in this study identified actionable items to inform the development and implementation of policies and programs that could bring the United States closer to HCV eradication.

## Supporting information

**S1 Checklist.**
(PDF)

**S1 Table. Codes for people who are in jail.** This is the codebook used for the interviews conducted with people who were incarcerated.
(PDF)

**S2 Table. Codes for non-incarcerated stakeholders.** This is the codebook used for the interviews conducted with non-incarcerated stakeholders.
(PDF)

**S1 File. Interview guide for people who are in jail.** These are the questions asked of participants who were incarcerated.
(DOCX)

**S2 File. Interview guide for non-incarcerated stakeholders.** These are the questions asked of participants who were not incarcerated.
(DOCX)

**S3 File. Interview guide for non-incarcerated stakeholders.** These are the questions asked of participants who were not incarcerated.
(DOCX)

## Acknowledgments

We are grateful for the support of the Middlesex Sheriff's Office in working with us to conduct research in their facilities and for their review of this work.

## Author Contributions

**Conceptualization:** Alysse G. Wurcel, Deirdre Burke, Tamsin A. Knox, Thomas Concannon, John B. Wong, Karen M. Freund, Curt G. Beckwith, Amy M. LeClair.

**Data curation:** Alysse G. Wurcel, Jessica Reyes, Julia Zubiago, Deirdre Burke, Tamsin A. Knox, John B. Wong, Karen M. Freund, Curt G. Beckwith, Amy M. LeClair.

**Formal analysis:** Alysse G. Wurcel, Jessica Reyes, Julia Zubiago, Deirdre Burke, Tamsin A. Knox, Amy M. LeClair.

**Funding acquisition:** Alysse G. Wurcel, Peter J. Koutoujian, Deirdre Burke, Tamsin A. Knox, Thomas Concannon, Stephenie C. Lemon, John B. Wong, Karen M. Freund, Amy M. LeClair.

**Investigation:** Alysse G. Wurcel, Julia Zubiago, Deirdre Burke, Tamsin A. Knox, Thomas Concannon, Curt G. Beckwith.

**Methodology:** Alysse G. Wurcel, Jessica Reyes, Deirdre Burke, Tamsin A. Knox, Thomas Concannon, Stephenie C. Lemon, Curt G. Beckwith, Amy M. LeClair.

**Project administration:** Alysse G. Wurcel, Jessica Reyes, Julia Zubiago, Peter J. Koutoujian, Deirdre Burke, Tamsin A. Knox, John B. Wong, Karen M. Freund.

**Resources:** Alysse G. Wurcel, Julia Zubiago, Deirdre Burke.

**Software:** Deirdre Burke.

**Supervision:** Alysse G. Wurcel, Julia Zubiago, Tamsin A. Knox, Thomas Concannon, Stephenie C. Lemon, John B. Wong, Karen M. Freund, Curt G. Beckwith, Amy M. LeClair.

**Validation:** Jessica Reyes, Julia Zubiago.

**Visualization:** Alysse G. Wurcel.

**Writing – original draft:** Alysse G. Wurcel, Thomas Concannon, Stephenie C. Lemon, John B. Wong, Karen M. Freund, Amy M. LeClair.

**Writing – review & editing:** Alysse G. Wurcel, Julia Zubiago, Peter J. Koutoujian, Deirdre Burke, Thomas Concannon, Stephenie C. Lemon, John B. Wong, Karen M. Freund, Curt G. Beckwith, Amy M. LeClair.

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
