## [Decision Letter · Decision Letter 0]

31 Dec 2020

PONE-D-20-35522

I’m not gonna be able to do anything about it, then what's the point?” A Broad Group of Stakeholders Identify Barriers to and Facilitators of HCV Testing in Massachusetts Jails

PLOS ONE

Dear Dr. Wurcel,

Thank you for submitting your manuscript to PLOS ONE. After careful consideration, we feel that it has merit but does not fully meet PLOS ONE’s publication criteria as it currently stands. Therefore, we invite you to submit a revised version of the manuscript that addresses the points raised during the review process.

I am enthusiastic about this manuscript and, while this is not a guarantee of publication of the revised manuscript, I am optimistic that the reviewers concerns can be addressed with minor changes. The bulk of their and my concerns were with the reporting of the methods, which can be addressed using the Equator Network guidelines for reporting of qualitative studies (COREQ checklist) as well as the other changes recommended below.

We look forward to receiving your revised manuscript.

Kind regards,

Andrea Knittel

Academic Editor

PLOS ONE

Additional Editor Comments:

I am in agreement with the reviewers regarding the minor changes needed to render the manuscript appropriate for publication. Please include a copy of the completed COREQ checklist from the Equator Network with your resubmission. This helps with methodological rigor and reporting.

Journal Requirements:

2. Please provide additional information regarding the considerations  made for the prisoners included in this study. For instance, please discuss whether participants were able to opt out of the study and whether individuals who did not participate receive the same treatment offered to participants.

Furthermore, please clarify how oral informed consent was documented.

Finally when reporting the results of qualitative research, we suggest consulting the COREQ guidelines: http://intqhc.oxfordjournals.org/content/19/6/349. In this case, please consider including more information about:  1) interviewers' training and characteristics ; 2) how participants were selected; 3) if a pilot study was tested; 4) if bias issues were considered. Moreover, please provide the interview guide used as supporting information.

4. Please amend your manuscript to include your abstract after the title page.

Reviewers' comments:

Reviewer's Responses to Questions

**Comments to the Author**

1. Is the manuscript technically sound, and do the data support the conclusions?

Reviewer #1: Yes

Reviewer #2: Yes

2. Has the statistical analysis been performed appropriately and rigorously? 

Reviewer #1: N/A

Reviewer #2: N/A

3. Have the authors made all data underlying the findings in their manuscript fully available?

Reviewer #1: No

Reviewer #2: No

4. Is the manuscript presented in an intelligible fashion and written in standard English?

Reviewer #1: Yes

Reviewer #2: Yes

5. Review Comments to the Author

Reviewer #1: Thank you for the opportunity to review this well written paper on an important topic. Here are specific comments to improve the paper.

-Table 1- Could you stratify out clinicians working inside and outside of jail? i.e. specify the number for each of those groups within the provider category?

-Methods- Please provide more detail regarding the methods, including how you identified study participants in each group. Please refer to the COREQ checklist (https://academic.oup.com/intqhc/article/19/6/349/1791966) for reporting and ensure you are reporting all relevant aspects of the study.

-Table 2- Why report these demographic characteristics for people in the jail and why not for other participants?

-page 5- In the first paragraph on the section on the stigma theme, you refer to a patient. Should this be participant? Or person in custody? I find this confusing.

-page 6- Should the title of the theme include both testing and treatment?

Reviewer #2: The paper is well written and the authors do a good job of laying out the significance and justification for the study. I applaud the team for being able to recruit and engage with so many different sectors of stakeholders concerning HIV testing and treatment in jails. I have some suggestions below to strengthen the manuscript.

Methods

• Describe in more detail the context for the interviews with persons who are incarcerated. Where were the interviews conducted? Was it a private setting? How did the researcher assistants ensure that potential participants gave fully voluntary and informed consent? Since an IRB defined “vulnerable” population was included in this study, I think the authors should be more transparent about this process for the reader.

• Similar to the point above, how were non-incarcerated stakeholders identified for approach?

• The authors state that two semi-structured interview guides were developed--one for incarcerated persons (“patients”) and one for other stakeholders--and discuss how they are similar, but now how they were different?

• Minor: similar to how initials were given for the interviewers, can the authors add the initials for the coders?

Findings

• For the theme “People who are incarcerated frequently did not know that HCV testing is offered in jail,” I think the authors should first state whether the Middlesex jail has a policy for testing or not. It seems like there is some ambiguity about testing on the part of jail providers (“One provider who worked in jail reported that HCV testing was not offered to everyone, while others said that it was offered at intake or sick call.”). I suggest discussing the HCV policy and practices first, and then the knowledge of the patients. It seems that in addition to not knowing that testing was availably, many patients do not know much about HCV, in general.

• This may be a personal preference, but I really dislike the headings in the findings. They are too long and too specific. I think that more general headings will better capture the themes that are captured in the data. For example, “stigma against people who use drugs is a barrier to HCV testing” can simply be “Stigma and Deservingness.” “Captivity and Transience” can be “Jails as Community Health Care Providers”. “Limited Financial Resources Prevents HCV Treatment” becomes Perceived Cost-Effectiveness, etc. The paper is focused specifically on HCV, but I think these themes can likely be generalized to the experience of incarcerated people and healthcare in jails more generally.

• The final section (Investment in Linkage to Care Programs) seems a little thin, especially compared to the other sections. This theme needs to be fleshed out more.

Discussion

• The authors did a good job of contextualizing their findings within the current scope of knowledge and literature about HCV testing and treatment. However, as a reader I kept waiting for the list of definitive recommendations. There are suggestions throughout, but I recommend adding a paragraph (either before or after the limitations) that has a clear statement of recommendations for HCV testing and treatment in jails.

• The authors write “The importance of engaging people who have been incarcerated to help develop HCV testing strategies cannot be overstated. Lived experience can help inform discussions about the best way to communicate preventative healthcare strategies with people who have low health literacy and may lack trust in clinicians working in the jail.” This is an important point that seems sort of tacked on. I suggest making this idea its’ own paragraph to draw out its importance and maybe discuss ways this has been implemented for other diseases/settings.

6. PLOS authors have the option to publish the peer review history of their article (what does this mean?). If published, this will include your full peer review and any attached files.

Reviewer #1: **Yes: **Fiona Kouyoumdjian

Reviewer #2: No

---

## [Author Response · Author response to Decision Letter 0]

6 Apr 2021

I have included responses in the cover letter.

---

## [Editor Report · Decision Letter 1]

16 Apr 2021

I’m not gonna be able to do anything about it, then what's the point?” A Broad Group of Stakeholders Identify Barriers to and Facilitators of HCV Testing in Massachusetts Jails

PONE-D-20-35522R1

Dear Dr. Wurcel,

We’re pleased to inform you that your manuscript has been judged scientifically suitable for publication and will be formally accepted for publication once it meets all outstanding technical requirements.

Kind regards,

Andrea Knittel

Academic Editor

PLOS ONE
---

## [Editor Report · Acceptance letter]

15 May 2021

PONE-D-20-35522R1 

“I’m not gonna be able to do anything about it, then what's the point?”: a broad group of stakeholders identify barriers and facilitators to HCV testing in a Massachusetts jail 

Dear Dr. Wurcel:

I'm pleased to inform you that your manuscript has been deemed suitable for publication in PLOS ONE. Congratulations! Your manuscript is now with our production department. 

Kind regards, 

on behalf of

Dr. Andrea Knittel 

Academic Editor

PLOS ONE